# Polymorphisms in Human APOBEC3H Differentially Regulate Ubiquitination and Antiviral Activity

**DOI:** 10.3390/v12040378

**Published:** 2020-03-30

**Authors:** Nicholas M. Chesarino, Michael Emerman

**Affiliations:** Divisions of Human Biology and Basic Sciences, Fred Hutchinson Cancer Research Center, Seattle, WA 98109, USA; nchesari@fredhutch.org

**Keywords:** APOBEC3H, ubiquitination, restriction factors, HIV

## Abstract

The APOBEC3 family of cytidine deaminases are an important part of the host innate immune defense against endogenous retroelements and retroviruses like Human Immunodeficiency Virus (HIV). APOBEC3H (A3H) is the most polymorphic of the human APOBEC3 genes, with four major haplotypes circulating in the population. Haplotype II is the only antivirally-active variant of A3H, while the majority of the population possess independently destabilizing polymorphisms present in haplotype I (R105G) and haplotypes III and IV (N15del). In this paper, we show that instability introduced by either polymorphism is positively correlated with degradative ubiquitination, while haplotype II is protected from this modification. Inhibiting ubiquitination by mutating all of the A3H lysines increased the expression of haplotypes III and IV, but these stabilized forms of haplotype III and IV had a strict nuclear localization, and did not incorporate into virions, nor exhibit antiviral activity. Fusion chimeras with haplotype II allowed for stabilization, cytoplasmic retention, and packaging of the N15del-containing haplotype III, but the haplotype III component of these chimeras was unable to restrict HIV-1 on its own. Thus, the evolutionary loss of A3H activity in many humans involves functional deficiencies independent of protein stability.

## 1. Introduction

Restriction factors are a critical component of the innate immune response to cellular infection by viruses by inhibiting various stages of the viral replication cycle. The APOBEC3 (A3) family of cytidine deaminases are potent restriction factors against endogenous retroelements and retroviruses, with four members of this family (A3D, A3F, A3G, and A3H) demonstrating considerable antiviral activity against lentiviruses such as Simian Immunodeficiency Virus (SIV) and Human Immunodeficiency Virus (HIV) [1]. The antiviral A3 proteins incorporate into nascent virions and act during the reverse transcription stage of retroviral infection to convert cytosines to uracils in viral single-stranded DNA [2]. The result of successful A3 activity is lethal hypermutation of the retroviral genome, preventing the establishment of a productive infection. Thus, the A3 proteins act in concert to provide a barrier to both cross-species transmission and within-host cell-to-cell spread of infection.

All of the *A3* genes display strong signatures of positive selection throughout primate evolution, suggesting a longstanding genetic conflict between this family of restriction factors and the pathogens and endogenous retroelements they restrict [3,4,5]. To overcome A3 restriction, lentiviruses have evolved the accessory protein, Vif, that potently inhibits the function of all antiviral A3 proteins simultaneously [6,7]. Vif hijacks a host cellular E3 ubiquitin ligase complex and acts as a substrate receptor for A3 proteins, promoting their ubiquitination and subsequent degradation via the proteasome [8,9,10,11]. The direct antagonistic relationship between host A3 and lentiviral Vif underlies an evolutionary arms race that selects for mutations in A3 that can escape Vif antagonism, and mutations in Vif that regain A3 binding [12,13]. Therefore, the A3 proteins confer selective pressure on lentiviruses to evolve Vif proteins that can overcome the host A3 defense.

Despite the selective pressure to retain host defense systems like the *A3* genes, restriction factors can become less effective due to circulating polymorphisms that reduce inherent antiviral function. It is possible that such polymorphisms may arise and become fixed in the population upon the loss of a selective pathogenic pressure, especially if the maintenance of a particular antiviral gene is deleterious to the host. A3H is a notable example of such a polymorphic restriction factor in humans, with four major A3H variants (haplotypes I, II, III, and IV) circulating in the human population, and at least eight other minor haplotypes identified to date [14]. Among these, haplotype II is the only variant to encode a protein with potent activity against lentiviruses [15,16,17,18,19]. In contrast, the protein encoded by haplotype I is far less antiviral and stable, and proteins encoded by haplotypes III and IV are not detectable at all [15,16,17,18,20,21]. Evolutionary evidence suggests that human A3H lost activity in two independent events since the last common ancestor with chimpanzees [16]. Thus, the destabilization of haplotypes I, III, and IV are a consequence of two independent single nucleotide polymorphisms, R105G in haplotype I and a deletion of codon 15, N15del, in haplotypes III and IV. Each mutation is sufficient to render A3H ineffective against HIV [15,16,18]. Importantly, the differences in protein expression are not explained by differences in transcript levels, as mRNA levels are comparable for both overexpressed stable and unstable alleles [22] as well as endogenous transcripts of various alleles measured in primary T lymphocytes [17]. Rather, the expression differences are due to reduced protein half-lives [16].

Recent structural studies of pig-tailed macaque [23], human [24,25], and chimpanzee [26] A3H have established a unique RNA interaction mechanism central to the antiviral function of A3H. Two A3H monomers interact with opposite sides of a short RNA duplex, and the A3H monomers in this complex interact primarily with the RNA duplex and not with one another. Importantly, mutagenesis of residues in the interaction interface between A3H and duplex RNA results in a decrease in A3H expression similar to the unstable human haplotypes [25,26,27]. Thus, the formation of the A3H-RNA complex may be a critical regulator of A3H stability. However, the exact mechanism in which R105G destabilizes A3H is not well understood, and even less is known of the effect of the N15del mutation in haplotypes III and IV.

In order to better understand the processes underlying the differential expression of different A3H haplotypes, we asked how ubiquitination, a posttranslational modification most widely known for its role in protein degradation, differs between A3H haplotypes. We found that the rates of ubiquitination were greater in unstable haplotypes I, III, and IV, while the stable haplotype II was largely protected from this modification. By genetically inhibiting ubiquitination through lysine mutagenesis, we were able to recover expression of the N15del-containing haplotypes III and IV to levels comparable to haplotype I. However, despite increasing protein expression, these changes did not restore antiviral activity of any of these haplotypes against HIV-1. Rather, stabilized versions of the proteins encoded by haplotypes III and IV were strictly localized to the nucleus and were unable to package into budding virions. Thus, the R105G and N15del mutations in A3H result in functional defects that cannot be restored by inhibiting ubiquitination alone. On the other hand, by fusing the stable and antiviral haplotype II to haplotype III, the deficiencies of the N15del mutation on expression, localization, and virion incorporation were reversed. However, we found that stabilized and packaged haplotype III was susceptible to processing by HIV-1 protease, and incapable of restricting HIV-1 when linked to an enzymatically inactive haplotype II. Taken together, these results suggest that both A3H stability and activity are tightly regulated by loss-of-function mutations, and selection for these mutations may serve to protect the host in the absence of a pathogenic pressure.

## 2. Materials and Methods

### 2.1. Plasmids

Plasmids containing A3H haplotypes I through IV were previously described [16], and used as templates to generate C-terminally HA-tagged constructs. All constructs used in this study were cloned into pcDNA3.1 (Thermo Fisher, Waltham, MA, USA) using EcoRI/XhoI restriction sites. Mutations were introduced using standard PCR or using the QuikChange Lightning Multi Site-Directed Mutagenesis Kit (Agilent, 210515, Santa Clara, CA, USA). Double A3H fusion constructs I–I, II–II, I–II, and II–I were previously described [20]. The double A3H fusion constructs unique to this study, III–III, III–II, and II–III were made through amplification of haplotypes II and III with a 5’ or 3’ primer containing the linker sequence GGT GGT GGT GGT GGC GCC (Gly-Gly-Gly-Gly-Gly-Ala). The 5’ and 3’ haplotype domains were digested using the KasI restriction site in the linker, and EcoRI (for 5’ domains) or XhoI (for 3’ domains). The 5’ domains, 3’ domains, and the EcoRI/XhoI double digested pcDNA3.1 vector were joined using T4 DNA Ligase (New England BioLabs, M0202, Ipswich, MA, USA). Infectivity experiments were performed using the HIV-1 molecular clone pLAIΔenvLuc2Δvif, which was previously described [28].

### 2.2. Cell Lines and Transfections

293T, HeLa, and SupT1 cells were obtained from ATCC (CRL-3216, CCL-2, and CRL-1942, respectively). The 293T and HeLa cells were cultured in DMEM, high glucose (Thermo Fisher, 11965092) supplemented with 10% HyClone Bovine Growth Serum (GE Healthcare Life Sciences, SH30541.03, Chicago, IL, USA) and 1× Penicillin-Streptomycin (Thermo Fisher, 15140122). SupT1 cells were cultured in RPMI-1640 (Thermo Fisher, 11875093) supplemented with 10% HyClone Fetal Bovine Serum (GE Healthcare Life Sciences, SH3091003) and 1× Penicillin-Streptomycin. Cell lines were mycoplasma free, as determined by the Fred Hutchinson Cancer Research Center Specimen Processing/Research Cell Bank core facility. Cells were incubated at 37 °C and 5% CO_2_ in a humidified incubator and maintained for under thirty passages before returning to a lower passage stock.

Transfections were performed using TransIT-LT1 Transfection Reagent (Mirus, MIR 2305, Madison, WI, USA) according to the manufacturer’s protocol. For Western blotting, 293T cells were plated at 1.5 × 10^5^/mL 24 h prior to transfection. Transfections in 12-well plates were performed using 0.5 µg/well of A3H plasmid and 3 µL/well of transfection reagent. For ubiquitination experiments, 6-well plates were transfected with 1.0 µg/well of A3H plasmid, 1.0 µg/well of myc-Ubiquitin, and 6 µL/well of transfection reagent. For infectivity experiments, 293T cells in 96-well plates (3.75 × 10^4^/well) were reverse-transfected with 60 ng/well pLAIΔenvLuc2Δvif, 30 ng/well A3H plasmid, and 10 ng/well L-VSV-G. For virion incorporation assays, transfections were performed in 6-well plates using 0.5 µg/well of A3H plasmid, 1.0 µg/well pLAIΔenvLuc2Δvif and 6 µL/well of transfection reagent. For immunofluorescence microscopy, HeLa cells were plated at 5.0 × 10^4^/mL on glass cover slips in a 12-well plate 24 h prior to transfection with 0.5 µg/mL of A3H plasmid and 3 µL/well of transfection reagent. In all cases, transfected cells were incubated for 48 h prior to downstream applications.

### 2.3. Viral Infectivity Assays

Viruses were propagated in 293T cells reverse-transfected with pLAIΔenvLuc2Δvif, A3H plasmid, and L-VSV-G for pseudotyping. Virus-containing supernatants were harvested 48 h after transfection, transferred to a V-bottom plate, and clarified of cells and debris by centrifugation at 1000× *g* for 3 min at 25 °C. An amount of 10 µL of supernatant was added to flat-bottom 96-well plates containing SupT1 cells (3.75 × 10^4^ and 90 µL/well) pretreated with 20 µg/mL DEAE/Dextran and mixed by repipetting. An amount of 5 µL of supernatant was saved for quantification of reverse transcriptase (RT) activity, as described previously [29]. Infected SupT1 cells were incubated for 48 h and lysed in 100 µL of Bright-Glo Luciferase Reagent (Promega, E2610, Madison, WI, USA). Infection was assessed by luciferase activity using a LUMIstar Omega microplate luminometer (BMG Labtech, Ortenberg, Germany), and raw luciferase values were normalized to 2000 mU RT activity.

### 2.4. Immunoprecipitation and Western Blotting

For experiments involving MG132, cells were treated with 10 µM MG132 (Selleckchem, S2619, Houston, TX, USA), or an equal volume of DMSO, in fresh media for 18 h after the initial 48 h transfection. For all other experiments, cells were harvested 48 h post-transfection. Cells were washed twice with PBS, and lysed on ice with RIPA buffer (50 mM Tris-HCl, pH 7.4, 150 mM NaCl, 1.0% Triton X-100, 0.5% sodium deoxycholate, 1 mM EDTA, 1 mM MgCl2) for 20 min. For ubiquitination assays, transfected cells were washed twice with PBS and lysed in 150 µL of 1% SDS buffer (50 mM Tris-HCl pH 7.4, 1.0% SDS) at 95 °C for 10 min. Lysate was passed through a 30-gauge needle to sheer DNA. An amount of 50 µg of lysate was saved for input. An equal concentration of lysate for immunoprecipitation was diluted in 100 µL of 1% SDS buffer, which was further diluted in 900 µL ice cold RIPA buffer lacking SDS containing 15 µL EZview Red Anti-HA Affinity Gel (Sigma-Adrich, E6779, St. Louis, MO, USA). Lysate was immunoprecipitated for 1 h at 4 °C with nutation, washed three times with ice cold RIPA buffer lacking SDS, and eluted in 40 µL 2× Laemmli Sample Buffer (BIO-RAD, 1610737, Hercules, CA, USA). Lysis buffers were supplemented with cOmplete Protease Inhibitor Cocktail (Roche, 11697498001, Basel, Switzerland), 10 µM MG132, and 50 µM PR-619 deubiquitinase inhibitor (Selleckchem, S7130). For virion incorporation assays, viral supernatants were passed through a 0.22 micron filter, and virus was pelleted at 21,000× *g* for 1 h at 4 °C then resuspended in 30 µL 4% SDS in PBS and 10 uL of NuPAGE 4× LDS Sample Buffer (Thermo Fisher, NP0007).

10 µg per lysate, 10 µL of immunoprecipitated sample, or 10 µL of viral lysate was resolved on a NuPAGE 4–12% Bis-Tris Protein Gel (Thermo Fisher, NP0336). Western blotting was performed using the primary antibodies anti-HA (Proteintech, 51064-2-AP, Rosemont, IL, USA), anti-myc (Proteintech, 16286-1-AP), anti-A3H (P1H6-1, [28]), anti-Lamin B1 (Proteintech, 66095-1-Ig), and anti-HIV-1 p24 (NIH AIDS Reagent Program, #3537, [30,31]) at a dilution of 1:2000. Secondary antibodies StarBright Blue 520 Goat Anti-Rabbit IgG (BIO-RAD, 12005869) and StarBright Blue 700 Goat Anti-Mouse IgG (BIO-RAD, 12005866) were used at a dilution of 1:10,000. Densitometric analysis was performed using ImageJ software [32]. All immunoblots are representative of at least three independent experiments.

### 2.5. Fluorescence Microscopy

Transfected cells on glass coverslips were washed twice with PBS and fixed in 4% paraformaldehyde, permeabilized in 0.1% Triton X-100 in PBS, and blocked in 2% Bovine Serum Albumin in PBS for 10 min each at room temperature. Primary antibodies anti-HA.11 (BioLegend, 901514, San Diego, CA, USA) or anti-A3H were used at a dilution at 1:500 and 1:50, respectively, in 0.1% Triton X-100 in PBS and incubated with cells for 20 min at room temperature. Cells were washed five times with 0.1% Triton X-100 in PBS prior to incubation with Alexa Fluor 488-labeled anti-mouse secondary antibody (Thermo Fisher, A-11001) at a 1:1000 dilution in 0.1% Triton X-100 in PBS. Cells were again washed five times prior to mounting glass coverslips in ProLong Gold antifade reagent containing DAPI (Thermo Fisher, P36935). Images were obtained using a Nikon E800 microscope at 40× magnification.

## 3. Results

### 3.1. Rates of Ubiquitination Differ among A3H Haplotypes

The majority of the human population possess *A3H* haplotypes that make unstable proteins with no antiviral activity. Haplotype I, containing the R105G mutation, has a global allele frequency of 46.4%, and haplotypes III and IV, containing N15del, make up a combined 30.2% of the population [14]. The protein encoded by *A3H* haplotype II has more steady-state expression than the protein encoded by haplotype I, and the proteins encoded by haplotypes III and IV are barely, if at all, detectable [15,16,17,18,33]. The question of why unstable and inactive variants of potent restriction factors occur at such high frequency in the human population and whether these inactive antiviral proteins could be manipulated to restore an innate immune function against HIV-1, prompted us to ask if A3H activity could be induced through stabilization, or if antiviral activity was lost independent of stability.

Previous work has shown that disruption of the proteasome with the drug MG132 modestly increases expression of both of the proteins encoded by *A3H* haplotypes I and II, suggesting that stability might be linked to proteasomal turnover [22,33]. We reexamined this question by evaluating the effect of MG132 on *A3H* haplotypes III and IV, which encode for the most poorly expressed proteins and contain the N15del polymorphism. We first constructed C-terminally HA-tagged A3H constructs (A3H-HA) and verified that the addition of this tag does not alter expression relative to their untagged counterparts (Appendix A). We then overexpressed each of the four HA-tagged A3H (A3H-HA) haplotypes in 293T cells, followed by treatment with the proteasomal inhibitor MG132 or a DMSO control. MG132 only slightly increased expression of haplotype II (1.2-fold). However, we saw a more dramatic effect of MG132 on the expression levels of unstable haplotypes I, III, and IV (3.7-, 5.6-, and 4.9-fold, respectively), suggesting that these haplotypes are more actively degraded via proteasomal degradation than haplotype II (Figure 1A).

The increased expression of all unstable A3H haplotypes following MG132 treatment suggests that these variants are naturally degraded to some extent through the ubiquitin proteasome system in the absence of the viral antagonist Vif. Because of the differential rates of recovery seen after MG132 treatment, we hypothesized that the rates of ubiquitination would similarly differ between protein variants. To this aim, we overexpressed each A3H-HA haplotype with myc-tagged Ubiquitin (myc-Ub). Interestingly, we noted that co-overexpression with myc-Ub increased the expression of unstable haplotypes to detectable levels, allowing us to probe for A3H ubiquitination without pretreating cells with MG132. A3H-HA was immunoprecipitated and subjected to Western blotting against myc-Ub. We observed higher amounts of ubiquitination for proteins encoded by haplotypes I, III and IV compared to haplotype II (Figure 1B, Appendix A: note ladder of Ub-containing proteins for haplotypes I, III, and IV). Importantly, myc-Ub bands in the immunoprecipitate were specific for A3H-HA, as a no A3H control did not carry over non-specific ubiquitin despite equal ubiquitination of substrates in the whole cell lysate (Appendix A). These results demonstrate that haplotypes I, III, and IV are processed by ubiquitination more heavily than haplotype II, which likely plays a large role in the turnover and loss of antiviral activity of the unstable haplotypes.

The finding that MG132 increases expression of haplotypes III and IV to levels at or exceeding that of haplotype I (Figure 1A) prompted us to investigate if restoring expression of N15del haplotypes would lead to an increase in antiviral activity. As MG132 has pleotropic effects and precludes reliable results from our infectivity assays, we employed the lysine mutagenesis approach previously used to study Vif-mediated polyubiquitination of A3G [34,35,36]. In order to prevent polyubiquitination of A3H, we converted each of the 14 lysines (K) in A3H to arginine (Figure 2A). By introducing arginine at these sites, we aimed to maintain the positively-charged regions thought to be important for RNA interaction and A3H dimerization [23,24,25,26]. Indeed, we found that transfection of the lysine-less mutants (indicated as “K-”) increased expression of proteins encoded by haplotypes III and IV (4.9-fold and 5.7-fold, respectively, Figure 2B). Importantly, when the lysines are mutated in any of the haplotypes ubiquitination is no longer observed (Figure 2C).

Interestingly, while mutation of lysines increased the expression of proteins encoded by A3H haplotypes III and IV, it decreased the expression of proteins encoded by haplotypes I and II (Figure 2B,C). These results were unexpected, as inhibition of ubiquitin-mediated proteasomal degradation does not lead to a similar decrease in expression of these haplotypes (Figure 1A). It is unlikely that these mutants are toxic to cells because we did not observe increased cell death upon transfection with each haplotype and their lysine-less mutants. Another possibility is that haplotypes I and II are actively stabilized by a mechanism that requires one or more of the lysine residues, perhaps due to ubiquitination, other modifications, or structural constraints. Regardless, it is important to note that lysine mutagenesis results in two opposing phenotypes between haplotypes I and II and haplotypes III and IV.

### 3.2. Inhibiting Ubiquitination Does Not Restore Antiviral Activity of Unstable A3H Haplotypes

We next assessed if increasing the expression of unstable A3H haplotypes III and IV by preventing all ubiquitination (Figure 2) could rescue antiviral function. To test this, we performed single-cycle infectivity assays by expressing wild-type A3H haplotypes, or their lysine-less counterparts, alongside *vif*-deficient HIV-1. Concurrent with their lower levels of expression (Figure 2B), the antiviral activities of lysine-less haplotype I and haplotype II were significantly reduced compared to their wild-type counterparts (2.1-fold and 5.9-fold loss in activity, respectively) (Figure 3A). Importantly, in the case of haplotype II, the lysine-less mutant is still capable of reducing infection to 11.1%, suggesting that neither lysine residues nor ubiquitination are required for the majority of anti-HIV-1 activity of A3H (Figure 3A), and this loss of function is likely due to the decreased expression of this mutant (Figure 2B). However, the increased expression of haplotypes III and IV induced by mutating all lysines to arginines (Figure 2B), did not similarly restore anti-HIV-1 activity (Figure 3A). Importantly, the antiviral activities of each mutant reflect the amount of packaging into virions (Figure 3B). As expected, wild-type haplotype II has the highest packaging efficiency, correlating with its expression and antiviral activity, while wild-type haplotype I has far less, but detectable, expression in virions. Lysine-less mutants of haplotypes I and II each lose some degree of viral packaging, with haplotype I losing all detectable expression (Figure 3B). In contrast, haplotypes III and IV are barely detectable in virions with or without lysine residues (Figure 3B). This result suggests that the N15del mutation in haplotypes III and IV inhibits antiviral activity in a mechanism that is independent of stability, but rather is linked to a defect that prevents virion packaging.

Previous studies have correlated cytoplasmic localization of A3 proteins with antiviral activity; for example, the less antiviral haplotype I has a nuclear-biased localization compared to the more cytoplasmic and active haplotype II [20]. However, due to the poor expression of wild-type haplotypes III and IV, we were unable to assess the contribution of the N15del mutation on A3H localization. By increasing expression of haplotypes III and IV through lysine mutagenesis, we reasoned we could use these mutants to determine the localization of N15del-containing variants. Importantly, the localization of lysine-less haplotype II was similar to its wild-type counterpart and was predominantly cytoplasmic, indicating that lysine mutagenesis does not alter the localization of A3H (Figure 3C). In contrast, immunofluorescence imaging of lysine-less haplotypes III and IV revealed a strong nuclear localization (Figure 3C). Taken together, these results suggest that the N15del mutation, like the R105G mutation in haplotype I [20], promotes localization of A3H to the nucleus, where it would be unable to package into nascent virions and elicit antiviral activity against HIV.

### 3.3. The Stability of A3H Haplotype II Is Dominant over the Instability of Haplotype III

The A3H haplotypes destabilized by the R105G (haplotype I) or N15del (haplotypes III and IV) mutations all share a nuclear-biased localization, a decrease in stability, and a decrease in antiviral activity. In a previous study, A3H constructs were made where haplotype II was linked to haplotype I, and was found that the chimeric proteins of haplotype I and II (I-II and II-I) were expressed at similar levels as haplotype II alone and much higher than haplotype I or a protein where haplotype I was duplicated (I-I) [20]. Therefore, the stability of haplotype II is dominant over the instability of haplotype I, suggesting that haplotype II is actively stabilized by a mechanism lost through R105G mutation. Taking a similar approach, we created double-domain A3H constructs to link haplotype II to haplotype III in both orientations (III-II and II-III) to address if the detrimental effects of the N15del mutation are dominant over haplotype II (Figure 4A).

To assess the effect of linking haplotype II to haplotype III on stability, we overexpressed these chimeras or wild-type A3H haplotypes and evaluated their expression by Western blotting. We anticipated one of two results: (1) the N15del mutation in haplotype III leads to active degradation even when linked to haplotype II, or (2) the active stabilization of haplotype II is dominant over the instability of haplotype III in a similar fashion to haplotype I. As shown previously, chimera I–I remains unstable, but haplotype I is stabilized (to levels similar to chimera II–II) when linked with haplotype II in either orientation (Figure 4B, [20]). Importantly, we note a similar increase in expression when haplotype III is linked to haplotype II in either orientation, but not when haplotype III is linked to itself, indicating that the stabilizing factors of haplotype II are also dominant over haplotype III (Figure 4B). Therefore, despite the more rapid turnover seen in haplotype III, association with haplotype II is sufficient to override this mechanism. The dominance of haplotype II stability over haplotype III suggests that the instability resulting from the N15del mutation is similar to that of the R105G mutation and is likely the result of passive ubiquitination and degradation rather than either of these mutations creating an active degradation signal. On the other hand, the ability of haplotype II to restore expression of both haplotypes I and III further supports the hypothesis that haplotype II stabilization is an active mechanism that correlates with protection from ubiquitination. Given that expression of the N15del-containing haplotype III can also be increased through linkage to haplotype II (Figure 4B), we next performed immunofluorescence microscopy to assess the localization of these chimeras. Imaging of the III–II and II–III chimeras showed a cell-wide distribution of chimeric A3H, demonstrating that cytoplasmic retention of haplotype II is also dominant over the nuclear localization of haplotype III (Figure 4C).

### 3.4. Stabilized Haplotype III Incorporated into Virions Is Unable to Restrict HIV-1

Because both the stability and cytoplasmic retention of haplotype II are dominant in these chimeras, we hypothesized that they also exhibit antiviral activity comparable to haplotype II. To this aim, we tested the ability of haplotype III chimeras to restrict *vif*-deficient HIV-1. As previously described, the antiviral activities of haplotypes I and II matched the respective activities of chimera I–I and II–II, and haplotype I linked to haplotype II (I–II and II–I) were comparably active to haplotype II (Figure 5A and [20]). We also found that linking haplotype III to itself (III–III), which does not rescue expression (Figure 4B), also does not considerably affect antiviral activity (Figure 5A). Finally, only linkage to haplotype II rescued antiviral activity of haplotype III against HIV-1 (Figure 5A). Interestingly, although both potently antiviral, the orientation of these chimeras had an impact on their restrictive capacity, as the III–II chimera was significantly less antiviral than chimera II–II and II–III (Figure 5A). Taken together, these results indicate that the stability, cytoplasmic retention, and antiviral potency of haplotype II are dominant over both destabilizing R105G and N15del mutations. Therefore, although we find additional functional deficiencies conferred by the N15del mutation in comparison to the R105G mutation (e.g., more nuclear localization, complete loss of packaging, and no antiviral activity), these deficiencies are similarly overcome through association with haplotype II. As association with haplotype II is able to similarly overcome both instability and functional defects of R105G and N15del mutations, it is likely that the downstream effect of these mutations is due to a shared loss of active stabilization and cytoplasmic retention central to the antiviral potency of haplotype II.

We found that viral incorporation of our chimeras matched their respective HIV-1 restriction with the surprising exception of haplotype III-containing chimeras (Figure 5B). While full-length chimeras III–II and II–III were not detected in viral lysates, we observed A3H staining below the expected molecular weight. The fact that chimeras III–II and II–III are stable in cells and antiviral strongly suggests that these chimeras are packaged at full-length but then processed by the HIV-1 protease. Interestingly, we also observed these suspected cleavage products in chimeras I–II and II–I but with a high amount of these full-length chimeras present (Figure 5B). Therefore, A3H chimeras may be prone to cleavage by HIV protease and are more or less vulnerable depending on the identity and orientation of the haplotypes in them.

Although the III–II and II–III chimeras exhibited considerable antiviral activity, the contribution of haplotype III to this phenotype remained unresolved. As these chimeras in particular were seemingly degraded within virions (Figure 5B), we suspected that only the haplotype II component was contributing to the majority of antiviral activity. In order to parse out the antiviral contributions of either haplotype in our chimeras, we mutated the active site of the haplotype II catalytic domain (E56A). We reasoned that the chimeras may require a single catalytically active domain for HIV restriction reminiscent of the other antiviral A3s, A3D, A3G, and A3F [6]. Single-cycle infectivity assays with these constructs revealed a striking loss of antiviral activity (13.9-fold between III–II and III–II_E56A_, and 14.3-fold between II–III and II_E56A_–III), suggesting that the active site of haplotype II is required for HIV-1 restriction when linked to haplotype III (Figure 5C). Importantly, this loss of activity was not concurrent with loss of expression (Figure 5D). Considering that haplotype III-containing chimeras lose activity when haplotype II catalytic activity is inhibited, as well as the increased susceptibility to these chimeras to HIV-1 protease, it is likely that haplotype III is largely inactive even when stabilized and packaged into virions.

## 4. Discussion

In the present study, we provide evidence that steady-state ubiquitination differs between the major A3H haplotypes circulating in the human population. These differences among A3H haplotypes in humans result in loss of protein expression that, only in the case of the N15del mutation, can be restored by mutation of lysines within the protein. However, the resultant protein is localized to the nucleus, fails to incorporate into virions, and does not have anti-HIV-1 activity. Our results indicate that the A3H antiviral activity was lost during human evolution, and that the inactive A3H proteins encoded by much of the human population cannot be easily restored by simply increasing the expression or by finding means to stabilize these proteins.

Ubiquitination most commonly results in proteasomal or lysosomal degradation of target proteins, although several other functions such as protein trafficking and inflammatory signaling have been described [37]. The function of ubiquitin modification on target proteins depends on the type of linkage (e.g., lysine-48 and lysine-64) and the particular lysine residues modified on the target protein. However, due to the partial recovery of unstable A3H haplotypes with the proteasomal inhibitor MG132 (Figure 1A), and the apparent lack of ubiquitination of the most stable A3H haplotype II (Figure 1B), it is likely that ubiquitination of haplotypes I, III, and IV primarily direct A3H for proteasomal degradation. It is important to note that each haplotype has three major splice variants that determine the length of the C-terminus, the most common two across all haplotypes being the 182 and 183 amino acid isoforms (SV182 and SV183, respectively) [14,15]. Interestingly, A3H haplotype II also expresses a 200 amino acid isoform at a frequency comparable to SV182 and SV183 [14]. While our study compares the ubiquitination of the SV183 isoform of all four haplotypes, how the ubiquitination of SV182 and SV200 may differ from SV183 is an interesting avenue of future research.

While we attempted to map the dominant lysine residues ubiquitinated in A3H, we were only able to see a loss of ubiquitination when all lysines were mutated. From this, we suspect that multiple lysines can be modified, and there is flexibility in lysine utilization in our more targeted lysine mutants. We found an intriguing phenotype by which mutating all lysines to arginine only increases expression of haplotypes III and IV and decreases expression of I and II (Figure 2B). The decrease in expression observed with lysine-less haplotypes I and II was unexpected, as treatment with MG132 resulted in an increase of these two haplotypes (Figure 1A). Therefore, it is possible that haplotypes I and II are regulated by ubiquitin in a manner independent from proteasomal degradation or that the presence of lysine residues are otherwise important for the stability of these haplotypes. Nonetheless, lysine-less A3H haplotype II is still potently antiviral in relation to its expression (Figure 3A) and maintains its cytoplasmic localization (Figure 3C). However, lysine-less haplotypes III and IV express detectable proteins that allowed us to assess known functionally relevant characteristics of A3H (Figure 2B). Lysine-less haplotypes III and IV are almost exclusively nuclear, are unable to package into virions and have no antiviral activity (Figure 3). In contrast, haplotype I has some cytoplasmic localization [20,38], and low but detectable viral packaging (Figure 3B) and antiviral activity (Figure 3A, Figure 5A). Taken together, these results reveal key differences between the R105G and N15del mutations in unstable A3H haplotypes, and suggest that the R105G mutation of haplotype I, although destabilizing, has more functional characteristics in common with the stable haplotype II than it does to the N15del-containing haplotypes III and IV.

Previous findings from our group demonstrated that the stability, cytoplasmic retention, and antiviral activity of haplotype II was dominant over the instability and nuclear bias of haplotype I by linking these haplotypes in either orientation [20]. The N15del mutation of haplotypes III and IV results in a protein that is less stable, more nuclear, and non-antiviral in comparison to haplotype I, suggesting that these haplotypes are more prone to negative regulation. Despite greater instability and functional deficiencies of N15del-containing haplotypes, we found that linking haplotype III to haplotype II in either orientation results in a stable, antiviral protein that localizes to the cytoplasm (Figure 4 and Figure 5). These findings suggest that A3H variants expressing the R105G and N15del mutations may share a common negative regulatory mechanism that affects the stability and localization of these mutants, and the N15del mutation leads to greater utilization of this mechanism.

We also observed an intriguing phenotype of our chimeric A3H proteins in that the packaging of full-length chimeras differs depending on the haplotypes linked. Unexpectedly, the stable and antiviral III–II and II–III chimeras were not detected at the expected molecular weight in Western blots of viral lysates, and only truncated A3H products were present (Figure 5B). There is precedent for HIV protease processing of one particular spliced isoform of A3H [14], and we suspect that these chimeras are more prone to cleavage by the HIV-1 protease than wild-type A3H. We also found truncated products of chimeras I–II and II–I in virions, but a large number of full-length chimeras were still present, suggesting that susceptibility to HIV-1 protease is also an important feature distinguishing the stable and unstable haplotypes (Figure 5B). Despite being stable, antivirally active, and presumably incorporated into virions, we found that the catalytic activity of haplotype II was required for the restriction capacity of haplotype III-containing chimeras (Figure 5C,D). This result suggests that the bulk of antiviral activity exhibited by chimeras III–II and II–III is conferred by haplotype II, and not from a stabilized and packaged haplotype III. These results indicate that the N15del mutation renders A3H nonfunctional even when stabilized and packaged into virions.

Several groups have recently described the mechanism by which stable A3H dimerizes through its interaction with duplex RNA, and this complex is maintained primarily through protein-RNA contacts [23,25,26,27]. Interaction with duplex RNA is mediated by aromatic and positively-charged residues in Loop 1, Loop 7, and the C-terminal α6 helix [23,25,26,27]. It is possible that the R105G and N15del mutations disrupt RNA binding and A3H-RNA complex formation, interfering with active stabilization and cytoplasmic retention observed with haplotype II. The R105G mutation is situated in β4 immediately upstream of the putative RNA-binding region (110-RLYYHW-115) in Loop 7 [39]. Although residue 105 is not directly implicated in RNA interaction, it is possible that mutations at R105 may alter the positioning of RNA-interacting residues in Loop 7. Importantly, the N15 residue is positioned within Loop 1, and deletion of this residue in haplotypes III and IV may directly prevent binding of A3H to its duplex RNA substrate and lead to greater instability and loss of function than what is conferred by the R105G mutation. Disrupting RNA binding prevents dimerization, and monomeric A3H may be susceptible to increased processing by ubiquitination. Indeed, mutagenesis of RNA binding residues results in decreased expression, nuclear localization, and a loss of packaging and antiviral activity [25,26,27,40,41], suggesting a shared mechanism of instability and loss of function to the R105G and N15del mutations. Recent biochemical work has more directly implicated nucleic acid binding as critical for A3H antiviral activity and is required for stabilization, virion incorporation, deaminase-independent inhibition of reverse transcriptase, and ssDNA substrate recognition [42]. Moreover, residues within Loop 1 play a dual role in both RNA binding and recognition of ssDNA, underscoring the importance of this region for the regulation of expression and antiviral function of A3H [42]. Thus, disruption of nucleic acid binding might explain all of the phenotypes observed for the unstable and inactive haplotypes I, III, and IV, as well as the functional differences observed between the R105G and N15del mutations.

Furthermore, the active mechanism by which haplotype II is stabilized might also be explained by RNA binding. Interestingly, five lysine residues are present in the C-terminal α6 helix in a span of 20 amino acids (Figure 2A), and ubiquitination of these residues may prevent duplex RNA binding. An intriguing model is that stabilization of haplotype II may result from a higher affinity of this haplotype for duplex RNA, which requires hydrogen bonding of arginine and lysine residues to RNA phosphates [23] and may preclude ubiquitination of these residues. While RNA-mediated dimerization may protect A3H from steady-state ubiquitination and degradation, recent work suggests that dimerized A3H is more susceptible to Vif-mediated degradation. HIV-1 Vif recognizes A3H in part due to its cytoplasmic localization resulting from RNA-mediated dimerization and binds to A3H in a manner that does not impact its interaction with RNA [41]. Thus, stable and active A3H may be protected from steady-state ubiquitination by nature of its unique RNA-mediated dimerization mechanism, while HIV-1 Vif has evolved to recognize and degrade the stabilized, cytoplasmic, and antiviral form of A3H.

Interestingly, the loss of A3H activity is not unique to humans, as an evolutionary analysis of A3H in Old World monkeys found that A3H activity has been lost in multiple primate species [43]. Moreover, several residues in the putative RNA binding region of Loop 1 were found to influence the antiviral potency of African green monkey and patas monkey A3H, providing further evidence that mutations introduced in this loop are a recurrent evolutionary mechanism that results in a loss of function [43]. It is unknown why these loss-of-function mutations of A3H arise and become fixed in multiple primate species despite the established importance of this restriction factor in the control of retroviruses. It is possible that the selective pressure to maintain A3H function has been lost due to the loss of an unknown pathogen or due to redundancy of more potent antiretroviral restriction factors such as A3G. On the other hand, functional A3H may come at a fitness cost in the absence of a pathogenic selective pressure. Although A3H haplotype I has lost much of its stability and antiviral activity, this variant has been associated with breast and lung cancer similar to the well-established cancer driver A3B [44,45]. Interestingly, a similar association has not been made for A3H haplotypes II, III, and IV, which may suggest a unique dysregulation of haplotype I promotes A3H-mediated genomic mutations driving these cancers. In contrast, the complete instability and loss of function coinciding with the N15del mutation of haplotypes III and IV may dually work to constrain a redundant restriction factor while protecting the host from genomic mutations. Thus, although unstable A3H haplotypes are all ineffective against HIV, a careful examination of these variants reveals differences in function that may have a significant influence on host fitness.

## Figures and Tables

**Figure 1 viruses-12-00378-f001:**
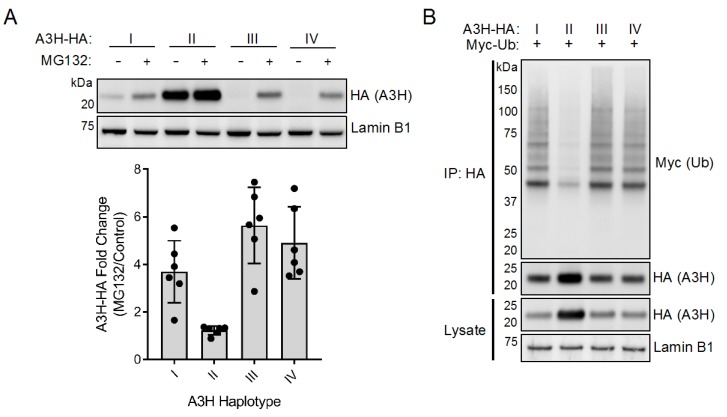
Human APOBEC3H (A3H) haplotypes are differentially ubiquitinated. (**A**) Top: Immunoblot of A3H haplotypes in the absence and presence of MG132. The 293T cells were transfected with plasmids expressing A3H-HA haplotypes I through IV, as indicated. Cells were then treated with MG132 (10 µM) or equal volume of DMSO for 18 h. Western blotting of whole cell lysates was performed using anti-HA to assess for A3H levels, and anti-Lamin B1 as a loading control. Bottom: Densitometric analysis of six independent experiments. Expression of A3H-HA was normalized to Lamin B1 levels and plotted as the fold change of MG132 over DMSO control. Error bars indicate standard deviation from the mean. (**B**) Immunoblot of ubiquitinated A3H haplotypes. The 293T cells were co-transfected with plasmids expressing A3H-HA haplotypes, as indicated, alongside myc-tagged ubiquitin (myc-Ub). Whole cell lysates were immunoprecipitated with anti-HA resin, and Western blotting was performed with anti-HA and anti-myc antibodies. Western blotting of whole cell lysates was performed using anti-HA to confirm expression of A3H, and anti-Lamin B1 as a loading control.

**Figure 2 viruses-12-00378-f002:**
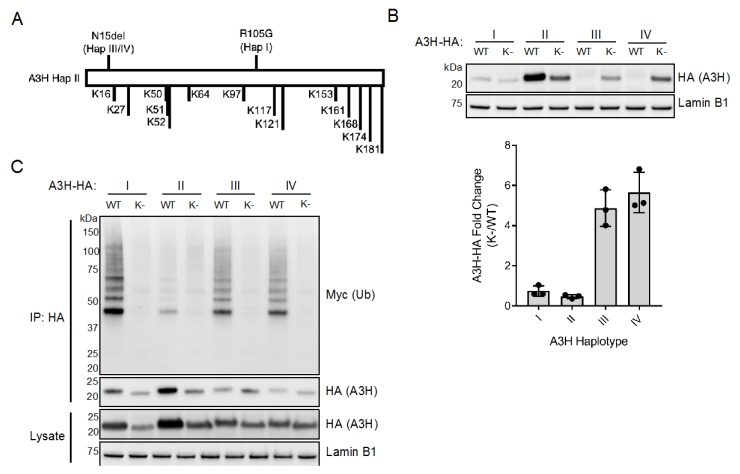
Genetic inhibition of ubiquitination increases expression of A3H haplotypes III and IV. (**A**) Protein schematic of A3H haplotype II (A3H Hap II) highlighting the polymorphic sites of haplotypes I (Hap I, R105G) and haplotypes III and IV (Hap III/IV, N15del), as well as all lysine residues (K) present in A3H. (**B**) Top: Immunoblot of wild-type (WT) and lysine-less (K-) A3H haplotypes expressed in 293T cells. Western blotting of whole cell lysates was performed using anti-HA to assess for A3H levels, and anti-Lamin B1 as a loading control. Bottom: Densitometric analysis of three independent experiments. Expression of A3H-HA was normalized to Lamin B1 levels and plotted as the fold change of lysine-less (K-) over wild-type (WT) haplotypes. Error bars indicate standard deviation from the mean. (**C**) Immunoblot of ubiquitinated wild-type (WT) and lysine-less (K-) A3H haplotypes co-transfected in 293T cells alongside myc-tagged ubiquitin (myc-Ub). Whole cell lysates were immunoprecipitated with anti-HA resin, and Western blotting was performed with anti-HA and anti-myc antibodies. Western blotting of whole cell lysates was performed using anti-HA to confirm expression of A3H, and anti-Lamin B1 as a loading control.

**Figure 3 viruses-12-00378-f003:**
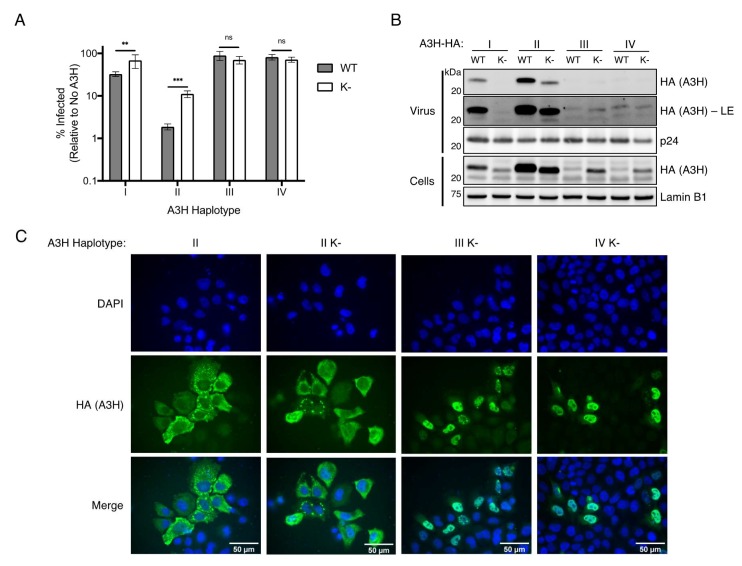
Functional deficiency of A3H haplotypes III and IV is independent of instability. (**A**) Single-cycle viral infectivity assays performed with *vif*-deficient Human Immunodeficiency Virus-1 (HIV-1) virus produced in the presence of wild-type (WT, gray bars) A3H haplotypes, or lysine-less (K-, open bars) counterparts. Shown is the average of six biological replicates with error bars representing the standard deviation of the mean. ** *p* < 0.01, *** *p* < 0.001, ns = not significant (*p* > 0.05), unpaired *t* test. (**B**) Immunoblot of viral supernatant (Virus, top) and whole cell lysate (Cells, bottom) of virus producing cells as in (**A**) using anti-HA to assess A3H levels packaged in virus or expressed in cells. Anti-p24 and anti-Lamin B1 were used as loading controls for virus and cell lysate, respectively. LE = long exposure. (**C**) Immunofluorescence imaging of HeLa cells transfected with wild-type A3H-HA haplotype II or lysine-less A3H-HA haplotype II (II K-), III (III K-), and IV (IV K-). Cells were stained for fluorescence microscopy using anti-HA to detect A3H (green), and DAPI to detect the nucleus (blue). Images are representative of 30 randomly selected field images across three independent experiments.

**Figure 4 viruses-12-00378-f004:**
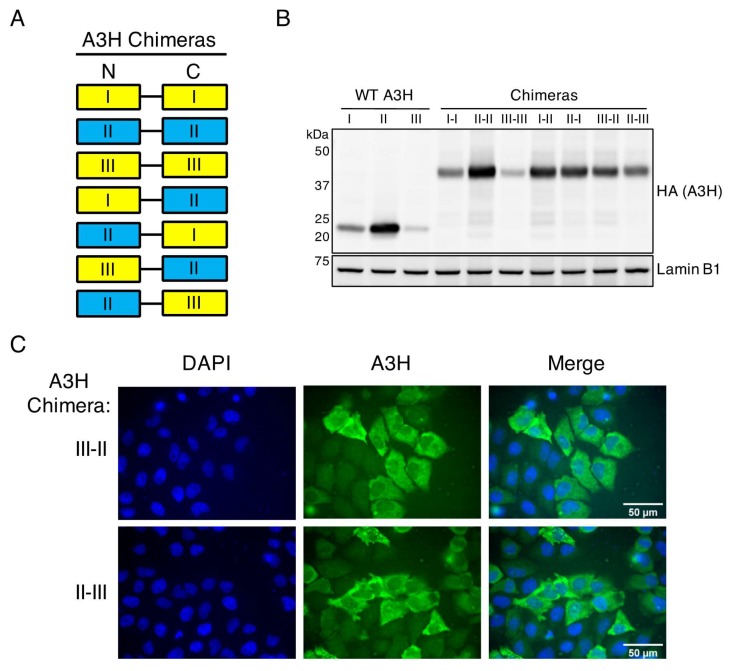
The stability and antiviral function of A3H haplotype II is dominant over the destabilizing N15del mutation of haplotype III. (**A**) Protein schematic of A3H chimeras used in this study. Haplotypes are denoted as I, II, or III in the N-terminal (N) or C-terminal (C) end of the flexible linker. Yellow boxes denote unstable haplotypes, and blue boxes represent active haplotypes. (**B**) Immunoblot of indicated wild-type (WT) A3H haplotypes or chimeras expressed in 293T cells. Western blotting of whole cell lysates was performed using anti-A3H and anti-Lamin B1 as a loading control. (**C**) Immunofluorescence imaging of HeLa cells transfected with A3H chimeras III–II, or II–III. Cells were stained for fluorescence microscopy using anti-A3H (green), and DAPI to detect the nucleus (blue). Images are representative of 30 randomly selected field images across three independent experiments.

**Figure 5 viruses-12-00378-f005:**
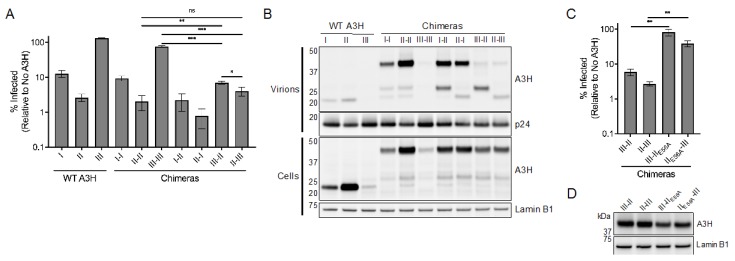
Stabilized and packaged A3H haplotype III is unable to restrict HIV-1. (**A**) Single-cycle infectivity assays performed with *vif*-deficient HIV-1 virus produced in the presence of the indicated wild-type A3H haplotypes or chimeras. Shown is the average of three biological replicates with error bars indicating the standard deviation of the mean. * *p* < 0.05, ** *p* < 0.01, *** *p* < 0.001, ns = not significant (*p* > 0.05), unpaired t test. (**B**) Immunoblot of viral supernatant (Virus, top) and whole cell lysate (Cells, bottom) of virus producing cells as in (**A**) using anti-A3H to assess levels of A3H packaged in virus or expressed in cells. Anti-p24 and anti-Lamin B1 were used as loading controls for virus and cell lysate, respectively. (**C**) Single-cycle infectivity assays performed with *vif*-deficient HIV-1 virus produced in the presence of the indicated A3H chimeras. Shown is the average of three biological replicates with error bars indicating the standard deviation of the mean. ** *p* < 0.01, unpaired *t* test. E56A is a mutation at the catalytic active site of hap II (**D**) Immunoblot of indicated A3H chimeras expressed in 293T cells. Western blotting of whole cell lysates was performed using anti-A3H and anti-Lamin B1 as a loading control.

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
