# Peer review of "Polymorphisms in Human APOBEC3H Differentially Regulate Ubiquitination and Antiviral Activity"

_viruses, 2020, doi:10.3390/v12040378_

Round 1
Reviewer 1 Report
The manuscript by Chesarino and Emerman investigates the molecular mechanism behind the differential stability of different naturally occurring haplotypes of antiviral enzyme APOBEC3H. First, they use biochemical (MG132) and genetic (K to R mutants) manipulations to probe the steady-state levels of these proteins. Interestingly, MG132 increases levels of all haplotypes with greatest effects on haplotypes III and IV but lysine mutagenesis only increased levels of haplotypes III and IV (and even decreased levels of haplotypes I and II). Second, they show that the increased expression of haplotypes III and IV does not restore anti-HIV-1 activity, likely because deltaN15 somehow compromises packaging into viral particles. Third, this negative result is explained in part by the deltaN15 haplotypes III and IV localizing to the nuclear compartment (ie. far away from sites of packaging on the plasma membrane). Finally, a panel of haplotype chimeras was constructed to assess phenotypic dominance with haplotype II prevailing. The results are consistent with both haplotype I and III/IV being loss of function mutations acquired independently in human ancestors.
Suggestions for improvement:
Fig 3 – presumably subcellular localization images of K-containing hapIII and IV proteins cannot be achieved due to low expression levels; if so, this should be stated explicitly (or representative images should be shown).
Fig 3 – does only haplotype III form nuclear aggregates, or haplotype IV as well? Image quantification here would be helpful.
Fig 4 – please show immunoblots for both cells and viral particles; readers will want to know if the antiviral phenotype tracks with packaging ability.
Fig 4 – it would be interesting to profile the G-to-A mutations in the viral cDNA resulting from these infections (and particularly from a hapII catalytic mutant fused to a hapIII or IV protein to assess whether the hapIII or IV proteins possess any ssDNA C-to-U catalytic activity or whether they are simply dead enzymes …)
Author Response
Thank you for taking the time to review our manuscript. Please find our point-by-point response to all three reviewers attached.

Reviewer 2 Report
Three out of four main A3H haplotypes do not show anti-HIV activity, and haplotypes III and IV are not even detectable at a protein level. Nevertheless, haplotype III is the second most frequent A3H haplotype in Africa and haplotype IV is the second most frequent haplotype outside Africa. This study is significant, because it provides data to explain why these haplotypes are not functional. This paper is very well written and all the experiments are solid. There are few minor points to address to strengthen this work:
1-As indicated in Ref 14 of the paper, A3H haplotypes are not identical in terms of the isoforms they produce. Specifically, A3H HapII has a different variant with an extended C terminus. Therefore comparing different A3H haplotypes using a single isoform (perhaps the one with 183 amino acids) may not be an apple to apple comparison. This is an important point because even viral packaging of different isoforms are different. No additional data are required for this paper, however, this point needs to be discussed.
2- What are the evidence that HA tags are not affecting A3H localization?
3- Is dimerization and localization of A3H affected when lysines are mutated?
Author Response

(The authors gave the same response as above.)

Reviewer 3 Report
The manuscript entitled “Polymorphisms in human APOBEC3H differentially regulate ubiquitination and antiviral activity” by Chesarino and Emerman investigated the mechanism of A3H protein stability, in relation to its ubiquitination state and antiviral activity. First of all, authors showed that A3H haplotypes are differentially expressed (Hap II >> HapI > Hap III and IV which are barely expressed) and ubiquitinated (Hap I, III and IV being the most compare to Hap II) in cells. By mutating all lysine residues of haplotype I (R105G) and haplotypes II and IV (N15del, deletion of codon 15), they showed that ubiquitination was indeed correlated with the low expression level of haplotypes III and IV, but not haplotypes I and II. Interestingly, the stabilized forms of Hap III and IV have a strict nuclear localization, thus explaining their lack of incorporation into viral particles and antiviral function. Chimeric domain fusion of Hap I/II and II/III suggest that the stability phenotype of Hap II is dominant over the instability phenotype of Hap I and III (restoration of the cytoplasmic localization and antiviral function). Finally, authors suggest that A3H stabilization may be driven by protein-RNA interactions (R105 is located close to loop 7 and N15 is positioned within loop 1, both loops are implicated with RNA binding) and that mutations of these haplotypes may serve to protect the host in the absence of a pathogenic pressure.
The manuscript of Chesarino and Emerman is addressing an interesting question on the most complex APOBEC3 protein. Indeed, A3H has many different haplotypes, different splice variants, which considerably complicates the study of this restriction factor. I appreciate the authors efforts to elucidate this issue and try to understand why some haplotypes have antiviral functions and some others not. This manuscript is well written, clearly structured and contributes considerably to the understanding of A3H haplotypes.
In my opinion, there are a few issues to address.
Major comments:
- lane 196: I would exclude haplotype II from this sentence as this haplotype is marginally increased in presence of MG132 (see figure 1A and AB).
- Figure 1B: a control lane is missing for the immunoprecipitation experiment (Myc-Ub without A3H-HA). Was MG132 used in this experiment? (not indicated in the legend). Same question for figure 2C.
- Figure 2: the expression profile of A3H haplotypes is different in panels B and C (lysates WT versus K-, and more pronounced for WT). Is this due to a difference of quantity of loaded material in the gel, or used to perform the immunoprecipitation? Please comment.
- It would have been interesting to mutate each lysine independently to see which one is the most important (or all of them). Have the author performed such experiments?
- Figure 3: please add A3H haplotype I K (-) in panel C. It is surprising to see it in panels A and B, and not in C. Have you tried to perform the same immunofluorescence study with WT haplotypes (I-IV) but in presence of MG132? In this case, what is the localization of A3H haplotypes?
- Figure 4: I would definitively show all A3H chimeras in panel C (I-I, II-II, III-III, I-II and II-I) and not only III-II and II-III. Please add more statistics in panel D (compare III-II and II-III with II-II for example, and I-II and II-I with I-I). Please the cite the work of Salamango et al., Mol. Cell. Biol. 2018 concerning A3H localization.
- In the discussion section, authors raise an interesting hypothesis concerning the RNA binding defect of haplotypes I, III and IV. Two recent papers on A3H-RNA interaction could be cited and discussed (Wang et al., J. Mol. Biol. 2019; Bohn et al., J. Virol. 2019). This is probably beyond the scope of the paper, but it would have been interesting to perform some biochemical assays and test this hypothesis.
Minor comments:
- lane 147: “transfected cells were and lysed”. A word is missing.
- lane 195: I think this is haplotype II (and not haplotype I).
- lane 259 and 262: figure 3 instead of figure 2.
Author Response

(The authors gave the same response as above.)

Round 2
Reviewer 3 Report
All comments have been taken into consideration